# Cognitive-behavioural therapy (CBT) for renal fatigue (BReF): a feasibility randomised-controlled trial of CBT for the management of fatigue in haemodialysis (HD) patients

Federica Picariello,[1] Rona Moss-Morris,[1] Iain C Macdougall,[2] Sam Norton,[1] Maria Da Silva-Gane,[3,4] Ken Farrington,[3,4] Hope Clayton,[3] Joseph Chilcot[1]

[1]Health Psychology Section, Psychology Department, Institute of Psychiatry, Psychology and Neuroscience, King's College London, London, UK
[2]Department of Renal Medicine, King's College Hospital, London, UK
[3]Department of Renal Medicine, Lister Hospital, Stevenage, UK
[4]University of Hertfordshire, Hertfordshire, UK

**Correspondence to**
Federica Picariello;
federica.picariello@kcl.ac.uk

## ABSTRACT

**Introduction** Fatigue is one of the most common and disabling symptoms in end-stage kidney disease, particularly among in-centre haemodialysis patients. This two-arm parallel group feasibility randomised controlled trial will determine whether a fully powered efficacy trial is achievable by examining the feasibility of recruitment, acceptability and potential benefits of a cognitive-behavioural therapy (CBT)-based intervention for fatigue among in-centre haemodialysis patients.

**Methods** We aim to recruit 40 adult patients undergoing in-centre haemodialysis at secondary care outpatient dialysis units, who meet clinical levels of fatigue. Patients will be randomised individually (using a 1:1 ratio) to either a 4–6 weeks' CBT-based intervention (intervention arm) or to a waiting-list control (control arm). The primary feasibility outcomes include descriptive data on numbers within each recruiting centre meeting eligibility criteria, rates of recruitment, numbers retained postrandomisation and treatment adherence. To assess the potential benefits of the cognitive-behavioural therapy for renal fatigue intervention, secondary self-report outcomes include measures of fatigue severity (Chalder Fatigue Questionnaire), fatigue-related functional impairment (Work and Social Adjustment Scale), sleep quality (Pittsburgh Sleep Quality Index), depression (Patient Health Questionnaire-9) and anxiety (Generalised Anxiety Disorder-7). Changes in fatigue perceptions (Brief Illness Perception Questionnaire), cognitive and behavioural responses to fatigue (Cognitive and Behavioural Responses to Symptoms Questionnaire), sleep hygiene behaviours (Sleep Hygiene Index) and physical activity (International Physical Activity Questionnaire–short form) will also be explored. These self-report measures will be collected at baseline and 3 months postrandomisation. Nested qualitative interviews will be conducted postintervention to explore the acceptability of the intervention and identify any areas in need of improvement. The statistician and assessor will be blinded to treatment allocation.

**Ethics and dissemination** A National Health Service (NHS) Research Ethics Committee approved the study. Any amendments to the protocol will be submitted to the NHS Committee and study sponsor.

**Trial registration number** ISRCTN91238019;Pre-results.

## Strengths and limitations of this study

► This is the first feasibility trial assessing the feasibility, acceptability and potential benefits of a psychological intervention for the management of fatigue among in-centre haemodialysis patients.
► The mixed-methods approach will help to evaluate comprehensively and in-depth the feasibility and acceptability of the cognitive-behavioural therapy for renal fatigue (BReF) intervention; the qualitative data will complement the quantitative findings and help to identify areas in need of improvement.
► The BReF intervention was developed systematically, using theory and evidence, with substantial input from patient and public representatives.
► As this is a feasibility trial, it is not powered to detect efficacy.

## INTRODUCTION

End-stage kidney disease (ESKD) is a chronic disease of the kidneys, characterised by inadequate renal functioning, where renal replacement therapy (RRT) is necessary to sustain life.[1] Haemodialysis (HD) is the most common RRT modality, filtering toxins out of the blood via an artificial extracorporeal blood circuit. A typical HD patient will be required to attend dialysis sessions three times a week for 3–4 hours each time.[2] On average, patients with renal disorder experience 14 symptoms, with fatigue emerging as one of the most persistent and debilitating symptoms.[3 4] Fatigue is a complex and subjective symptom characterised by extreme and persistent tiredness resistant to rest and recuperation.[5–7] Forty-nine to 92% of dialysis patients suffer from fatigue.[3] Fatigue is a substantial contributor to impaired functioning and quality of life, and recent evidence suggests that it has also implications for clinical outcomes[8–14];

yet, it is often under-recognised and undertreated by healthcare professionals, perceived as a normal consequence of the illness and treatment burden.[15 16]

Current management in the form of pharmacological treatments or exercise is ineffective[17 18] and no theory-driven and evidence-based psychological interventions aimed at fatigue for this group currently exist, although there is some promising evidence for some improvements in fatigue following psychological interventions not aimed at fatigue specifically.[19] As the aetiology of fatigue in patients with renal disorder is still largely unknown, no consistent treatment model exists.[8 20]

There is increasing recognition regarding the importance of psychological factors in the perpetuation and maintenance of fatigue in other long-term physical conditions.[21–26] For example, in multiple sclerosis (MS), negative fatigue beliefs, such as catastrophising about the consequences of fatigue, embarrassment about fatigue and belief that fatigue is a sign of physical damage; and unhelpful behaviours in response to fatigue, like excessive resting or overdoing things followed by long resting periods to recover; were found to be strongly associated with fatigue, above and beyond the role of demographic and clinical factors, such as neurological impairment and remission status.[22] On the other hand, social support may act as a buffer to fatigue perpetuation.[27] An understanding of the contribution of these factors to fatigue has translated into successful psychological interventions, leading to clinically significant improvements in fatigue severity and fatigue-related functional impairment.[28 29] There is, in fact, extensive evidence in support of cognitive-behavioural therapy (CBT) for adjustment and the management of symptoms, like fatigue, in the context of long-term physical conditions, such as cancer and MS,[30 31] despite being originally developed for the treatment of mood disorders.[32 33] CBT is a structured, tailored and time-limited talking therapy that focuses on changing negative beliefs and unhelpful behaviours, as well as relaxation techniques, stress management and mindfulness to foster resilience.[33 34]

We conducted preliminary work, consisting of prospective and qualitative studies, which revealed the importance of cognitive and behavioural factors in the experience of fatigue in ESKD, in line with the findings from other long-term physical conditions. To date, the effectiveness of CBT specifically for fatigue has not been examined in this population, yet, a similar approach may also be useful here. Based on the findings from these precursor studies and patient and public involvement (PPI) input, we adapted an existing CBT approach initially developed by one of the authors (RMM) for fatigue in MS. According to this renal fatigue treatment formulation, which integrates biological and psychosocial factors; there are a number of factors that may act as triggers of fatigue in this patient population, such as anaemia and HD. While these factors trigger initial symptoms of fatigue, one's thoughts, emotions and behaviours in response to fatigue may maintain and perpetuate fatigue, potentially leading to a vicious cycle of negative illness and fatigue beliefs, increased distress and maladaptive behaviours, as displayed in figure 1. The factors maintaining and perpetuating fatigue are targeted in CBT.

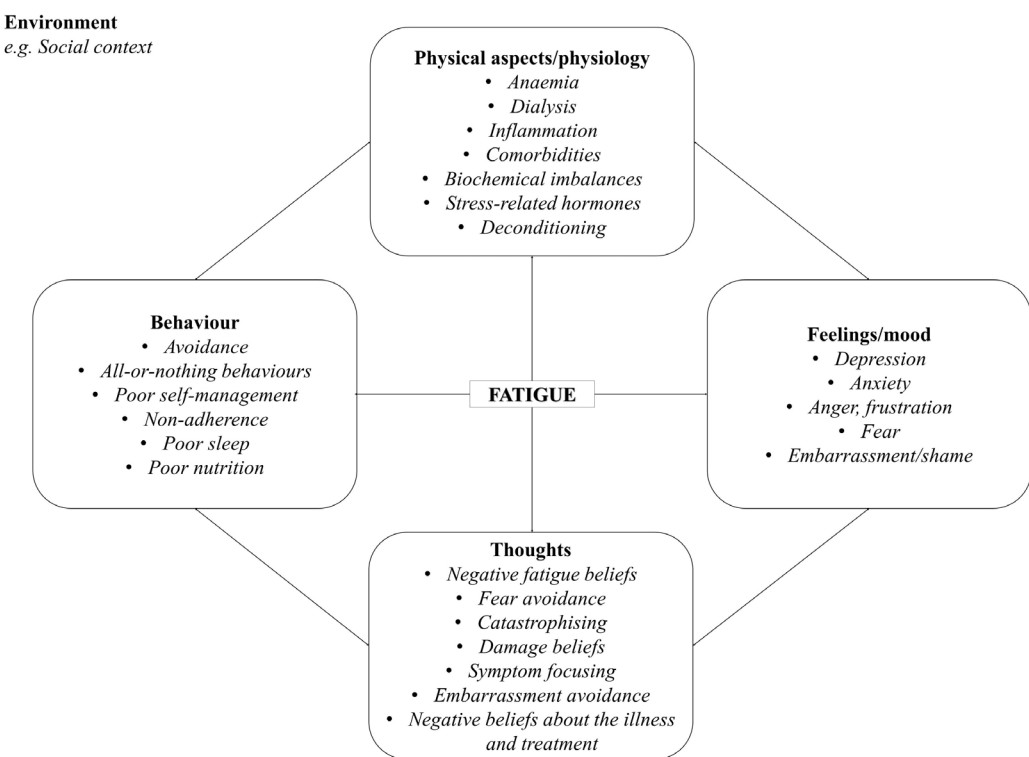

**Figure 1.** Cognitive-behavioural therapy model of renal fatigue. This diagram illustrates the different clinical, social situational and psychological factors that contribute to fatigue in this setting.

## Objectives

A feasibility design was deemed necessary to determine whether a full-scale randomised controlled trial (RCT) is feasible, by considering numbers meeting eligibility criteria, rates of recruitment and retention postrandomisation, floor/ceiling effects that might affect sensitivity to change; as well as by identifying any intervention-specific issues, particularly occurrence of adverse events and timing burden. The following aims will be addressed in this feasibility RCT:

▶ Objective 1: to estimate rates of recruitment and retention.
▶ Objective 2: to estimate willingness to be randomised.
▶ Objective 3: to explore the level of adherence to the intervention (intervention arm only).
▶ Objective 4: to estimate the standard deviation (SD) of fatigue in this patient population in order to compute a more robust estimate of the sample size required for an efficacy trial.
▶ Objective 5: to preliminary assess the psychometric properties of the self-report instruments used.
▶ Objective 6: to explore the potential benefits of the intervention at reducing fatigue severity and fatigue-related functional impairment as compared with the waiting-list control.
▶ Objective 7: to explore the potential benefits of the intervention at reducing depression and anxiety and improving sleep quality as compared with the waiting-list control.
▶ Objective 8: to examine change in fatigue-related cognitions and behaviours, and whether their effect differs between the intervention and control arm.
▶ Objective 9: to qualitatively explore patient perceptions of the acceptability and usefulness of the intervention and identify areas of improvement for a future full-scale trial.
▶ Objective 10: to explore any intervention-specific issues, particularly setting, mode of delivery of the intervention and acceptable number of sessions/chapters.

## METHODS

### Design

A two-arm parallel group feasibility RCT. There will be one follow-up assessment at 3-month postrandomisation. A nested-qualitative study will evaluate patients' experiences with the intervention.

### Setting and participants

Outpatient HD patients will be recruited from two National Health Service (NHS) sites in England.

### Inclusion and exclusion criteria

Participants are eligible for the study if they:
▶ are >18 years of age
▶ have a confirmed ESKD diagnosis
▶ are experiencing clinical levels of fatigue defined as scoring >18 on the Chalder Fatigue Questionnaire (CFQ), when using the continuous scoring[35 36]
▶ have full verbal and written proficiency in English
▶ are receiving in-centre HD
▶ length of time on dialysis >90 days
▶ are willing and able to take part in the study and intervention.

Patients will be excluded if they:
▶ do not provide informed consent or refuse to be randomised
▶ have any known cognitive impairments
▶ have a severe mental health disorder, for example, psychosis and bipolar disorder
▶ do not have full verbal and written proficiency in English
▶ are currently receiving psychotherapy
▶ are currently participating in any other intervention trial
▶ are failing on dialysis and approaching end of life (supportive care/palliative care pathway)
▶ have a fatigue (CFQ) score below the cut-off at the prerandomisation assessment (spontaneous improvement after screening).

Patients will not be screened for anaemia. Levels of haemoglobin and haematocrit are generally maintained within recommended ranges.[17] Additionally, there is evidence for a ceiling effect of anaemia management on fatigue[37] and improvements in fatigue are often below a clinically meaningful threshold, particularly in patients on dialysis.[38] Nonetheless, the role of anaemia-related factors will be examined in the exploratory analysis and may lead to changes to the inclusion/exclusion criteria for a future efficacy trial.

### Flow of recruitment and participant timeline

Recruitment will take place from October 2017 to July 2018. Patients interested in participating will be given a participant information sheet, screening questionnaire, consisting of sociodemographic and illness-related questions and the CFQ; an informed consent form and a freepost envelope. Potential participants will be given a minimum of 24 hours to establish if they would like to take part. Following consent, eligible participants will receive the baseline questionnaire, and will be randomised after the completion of the baseline questionnaire. Participants who score below the fatigue cut-off at the prerandomisation assessment will be excluded.

Participants will be informed of the outcome of the randomisation process over the phone and will receive confirmation of their treatment allocation and materials via post. We anticipate the participant's journey through the study will last approximately 4–5 months, as summarised in figure 2, with approximately 1 month dedicated to screening and randomisation. The intervention will last between 4 and 6 weeks, depending on each participant's needs. Participants are expected to complete one session per week. Follow-up data will be collected at 3 months postrandomisation (T1). On completion

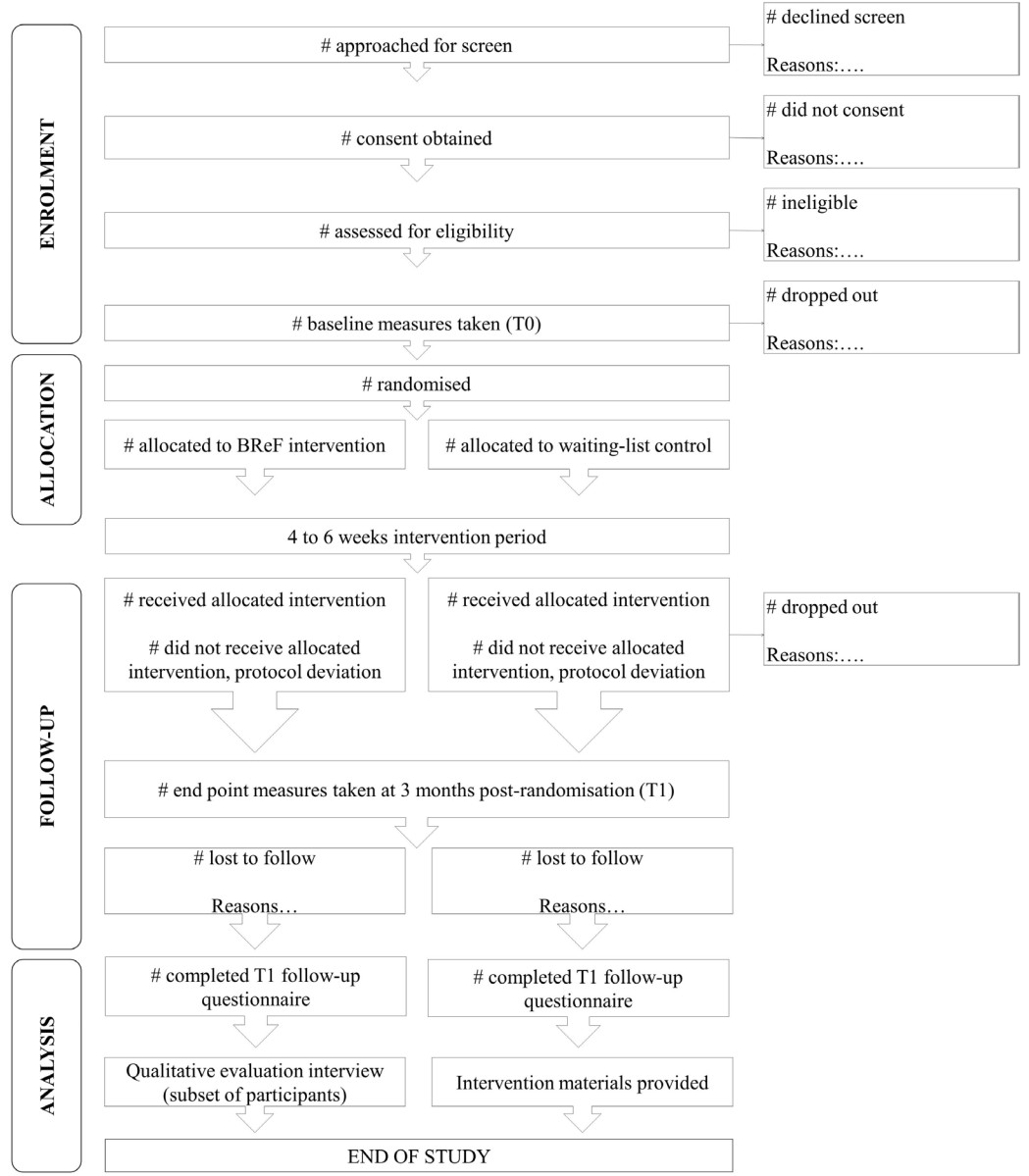

**Figure 2** Anticipated flow of participants through the study. Number of patients approached for screen, those who consented, and those who were assessed for eligibility will be recorded. Eligible patients will be invited to complete a baseline questionnaire (T0). After completion of the baseline questionnaire, participants will be randomised. Participants in the intervention arm will receive the intervention over 4–6 weeks. All participants will complete a follow-up questionnaire at 3 months postrandomisation (T1). Participants in the intervention arm will be invited to take part in a qualitative evaluation interview at the end of their involvement in the study. After completion of the follow-up questionnaire, participants in the control arm will receive the intervention materials.

of the postintervention questionnaire, a subsample of participants will be invited to take part in the qualitative interview. After completion of follow-up assessments at T1, participants in the control condition will receive the intervention manual and tasks workbook via post.

## Randomisation, allocation concealment and blinding

Participants will be randomised in a 1:1 ratio to receive either the cognitive-behavioural therapy for renal fatigue (BReF) intervention or to the waiting-list control. Participants will be randomised at the individual level. Randomisation will be stratified by centre and randomly varying

block sizes will be used to maintain balance of numbers in each arm across the period of recruitment while maintaining allocation concealment. King's College London's independent randomisation service will be used. Because the randomisation sequence is automated in real time, the allocation sequence is concealed from researchers. The trial coordinator will receive an automated email with the outcome of the randomisation procedure. The nature of the trial is such that blinding of participants cannot be achieved. Follow-up outcomes will be completed independently by participants by post or online. If baseline or follow-up questionnaires are not completed, then

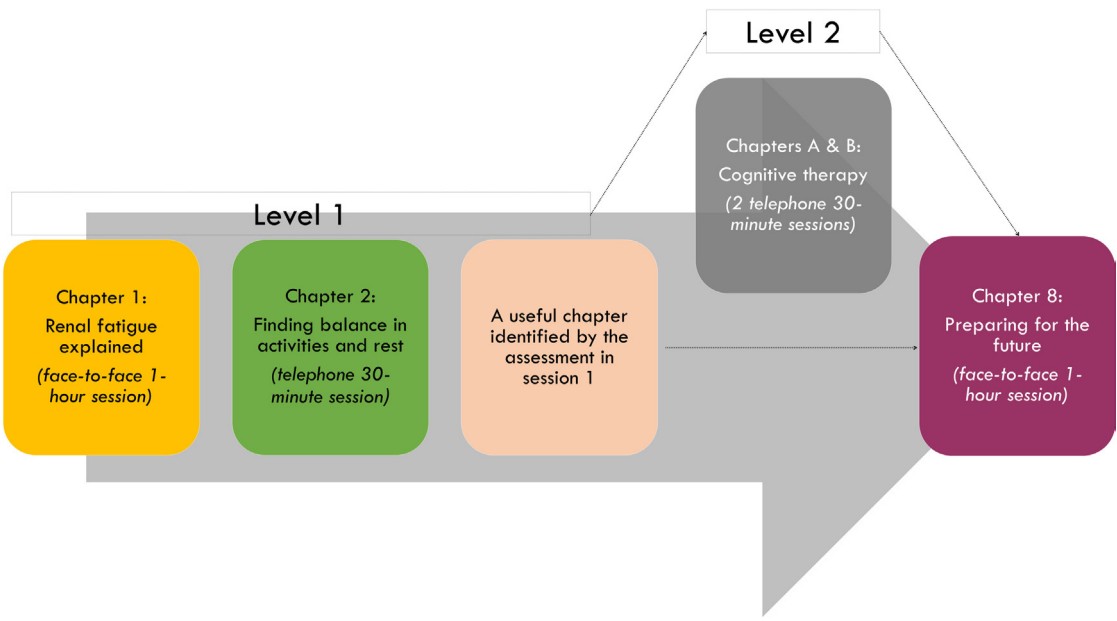

**Figure 3** Structure and content of the intervention. The intervention will follow a stepped approach over 4–6 weeks, accompanied by 3–5 sessions with a therapist. In level 1, participants will cover chapters 1, 2, a relevant chapter identified in the assessment and chapter 8. Level 2 focuses on cognitive therapy.

participants will receive reminder emails or phone calls and an assistance-based visit at the dialysis unit, by an independent researcher, who has not been involved in the intervention delivery. The statistician will remain blinded to treatment allocation.

### BReF intervention

The intervention is a tailored CBT-based self-management intervention with therapist support. The purpose of this intervention is to target individuals' fatigue beliefs and behaviours in order to facilitate coping with renal fatigue (ReF). Further detail on the intervention can be found in figure 3. Therapist guidance was deemed necessary to facilitate engagement with the programme, particularly in the formulation of a personal biopsychosocial model of fatigue and identification of unhelpful thoughts and behaviours.[39–41]

The development of the CBT-based intervention was systematic, based on the findings of reviews and qualitative and prospective studies, with substantial input from 10 patient and public representatives and a multidisciplinary team of health psychologists, clinical psychologists and nephrologists. The structure and content of the manual was drafted based on previous CBT interventions developed by one of our authors (RMM), such as '*Managing Your Multiple Sclerosis Fatigue: A Cognitive behavioural therapy manual*' and '*Improving Distress in Dialysis*'[42–46] and other sources.[47–49]

Participants will be provided with a structured CBT manual and a tasks workbook, including goal-setting sheets. This will be accompanied by three to five sessions with either a primary researcher who has a background in health psychology, basic CBT training and experience in working with fatigued patient groups (FP) or a registered health psychologist working in the renal setting (HC). In accordance with CBT principles, participants will be encouraged to complete tasks between sessions. Completion of these tasks has been found to be predictive of CBT outcomes.[50 51]

The manual consists of 10 chapters, accompanied by a tasks workbook for each session. Please see table 1 for the content of each chapter and associated tasks to be completed between sessions. The programme consists of two units, the basic unit (level 1) or the advanced unit (level 2). In the basic unit, participants will cover four chapters out of the manual, three will be accompanied by therapist sessions and one will be selected according to participants' needs, established in the assessment. Participants, engaging well in the first two sessions, will be given the opportunity to cover an additional two chapters with the therapist over the phone (level 2). Engagement will be discussed in clinical supervision, and will be assessed through, for example, completion of between session tasks and focus maintained during the sessions. The brief and stepped structure of the intervention was chosen to meet patients' needs, particularly their fatigability and potential concentration difficulties, according to the practical considerations previously raised with regards to the delivery of CBT in MS.[52] Sessions will be guided by participants' needs, identified through the self-monitoring tasks.

The first and last sessions will be face-to-face lasting 1 hour, while the remaining sessions will be over the phone, and will last 30 min. A combination of face-to-face and telephone sessions has been previously suggested for CBT in MS, to overcome possible limitations of individual delivery methods.[52] Sessions will be scheduled at

**Table 1** Summary of the content of the BReF manual

| Chapter | Content | Between sessions task |
|---|---|---|
| ReF explained | Understanding ReF and alternative explanations<br>A model for ReF<br>Assessment of fatigue | Fatigue self-monitoring |
| Finding balance in activities and rest | Patterns of rest and activity and its effects on the body<br>Planning activity and rest<br>Exercise | Activity difficulty task<br>Activity and rest goal sheet |
| Improving sleep | Sleep hygiene<br>Maladaptive sleep patterns<br>Improving sleep | Sleep, activity and rest goal sheet |
| Learning to relax | Diaphragmatic breathing<br>Progressive Muscle Relaxation (PMR)<br>Relaxation training: step-by-step | Relaxation diary |
| Coping with emotions | Strategies to cope with negative emotions<br>Self-assessment of negative emotions<br>Expressing emotions | Coping with negative emotions goal sheet |
| Managing stress | General tips to reduce the impact stress has on your life<br>Managing controllable and uncontrollable stressors<br>Mindfulness | Managing stress goal sheet |
| Making use of your social support | Creating a support network<br>Disclosure versus keeping it to self<br>Social comparisons | Social support goal sheet |
| Becoming aware of your thinking | Common unhelpful thoughts<br>Identifying unhelpful thinking | Thought record |
| Changing your thinking | Identifying alternative thoughts | Alternative thoughts goal sheet |
| Preparing for the future | Sustaining and building on improvements<br>Developing future goals<br>Tips for everyday life | Long-term goals worksheet |

BReF, cognitive-behavioural therapy for renal fatigue; PMR, progressive muscle relaxation; ReF, renal fatigue.

times that suit the participants. Face-to-face sessions will be conducted in a private environment. For the telephone sessions, participants will be encouraged to have the sessions in a quiet and private environment and allocate sufficient time not to feel rushed. Please see the Template for Intervention Description and Replication (TIDieR)checklist for a summary (table 2).

### Waiting-list control (control arm)
Participants allocated to the control arm of the study will receive their usual renal care, consisting of attending dialysis. As part of this feasibility trial, what constitutes usual care will be monitored to determine a control arm for a future efficacy trial and to handle potential contamination between the arms. After completion of the follow-up questionnaire, participants in the control group will receive the intervention manual and tasks workbook, but will not receive therapist support sessions. To minimise attrition from the control group, participants will be called to remind them that they can gain access to the programme in the weeks that follow.

### Clinical supervision
FP and HC will receive training on how to deliver the therapist support sessions from RMM, with approximately 3–4 hours of face-to-face contact, in addition to audio-recorded role-playing sessions with feedback. FP and HC will receive continuous supervision throughout the intervention from RMM, following the framework developed to support the delivery of psychological therapies with persistent physical conditions.[53] This will involve reflection and discussion of the sessions, feedback on the audio-recorded sessions and case management, particularly following the initial session with each participant in terms of the treatment plan and subsequently discussions around each participant's progress over the course of the intervention and progression to the level 2 sessions.

### Intervention fidelity
Two therapists will deliver all the intervention sessions following the detailed and structured manual developed for the patients. With permission from the participants, sought on the consent form, therapy sessions will be audio-recorded and a random sample assessed for fidelity during supervision by RMM.

### Data collection and feasibility outcomes
The primary focus of this trial is the feasibility of the BReF intervention.

### Primary feasibility outcomes
Feasibility will be assessed by collecting descriptive data on recruitment and retention rates and willingness to be randomised according to Consolidated Standards of Reporting Trials feasibility and pilot trial guidelines.[54] In the intervention arm, the degree of adherence to the intervention will also be assessed by recording completion of the chapter selected in session 1 and completed by participants independently in week 3 and the between sessions tasks, as well as recording of attendance at therapist sessions (ie, did not attend (DNA's) and adherence to the assigned session time). Uptake of and adherence

**Table 2** TIDieR checklist summary

| | Item 1: brief name | Item 2: rationale | Item 3: materials | Item 4: procedure | Item 5: who provided | Item 6: delivery mode | Item 7: location/ setting | Item 8: when and how much | Item 9: tailoring | Item 10: how well (planned) |
|---|---|---|---|---|---|---|---|---|---|---|
| BReF | Please see the Introduction and intervention description (section BReF intervention) | Please see the Introduction and intervention description (section BReF intervention) | BReF manual and tasks workbook | Please see sections: Flow of recruitment and participant timeline and BReF intervention | Primary researcher who has a background in health psychology and experience in working with fatigued patient groups or registered health psychologist working in the renal setting. | All sessions individual: –Two sessions face-to-face –One to three sessions over the phone | Recruitment from outpatient HD units in the UK. Therapy sessions in a private environment. | 3–5 weekly sessions with the therapist, depending on engagement. Two sessions face-to-face, lasting 60 min. 1–3 sessions over the phone, lasting 30 min. 1 session completed independently. | Yes, tailored: optional session determined through the assessment in session 1. Tailored to participants' needs, identified through the self-monitoring. | Therapists will follow a structured intervention manual. Therapy sessions will be audio-recorded and assessed for fidelity by the supervisor, RMM. |

BReF, cognitive-behavioural therapy for renal fatigue; HD, haemodialysis.

**Table 3** Schedule of assessments

| | Time | | |
|---|---|---|---|
| | | **Baseline** | **Postintervention** |
| **Assessment** | **Screening** | **(T0)** | **(T1)** |
| CFQ | x | x | x |
| WSAS | | x | x |
| PHQ-9 | | x | x |
| GAD-7 | | x | x |
| PSQI | | x | x |
| BIPQ | | x | x |
| CBSQ | | x | x |
| SHI | | x | x |
| IPAQ-short | | x | x |
| CCI | x | | |
| Sociodemographic characteristics | x | x | |
| Clinical characteristics | x | x | |
| Biochemical outcomes | | x | x |
| Self-reported adverse events | | | x |
| Self-reported treatments for distress or fatigue during study | | | x |
| Qualitative interviews | | | x |

BIPQ, Brief Illness Perceptions Questionnaire; CBSQ, Cognitive and Behavioural Responses to Symptoms Questionnaire; CCI, Charlson Comorbidity Index; CFQ, Chalder Fatigue Questionnaire; GAD-7, Generalised Anxiety Disorder-7; IPAQ-short, International Physical Activity Questionnaire -short form; PHQ-9, Patient Health Questionnaire-9; PSQI, Pittsburgh Sleep Quality Index; SHI, Sleep Hygiene Index; WSAS, Work and Social Adjustment Scale.

to the level 2 sessions will also be recorded. Given the exploratory nature of this trial, the number of completed intervention components will be assessed, this may help to identify an adherence cut-off for a future efficacy trial.

### Secondary self-reported patient outcomes

Self-reported patient outcomes will also be collected at baseline (T0) and 3 months postrandomisation (T1) via post or online. The assessment schedule completed by patients is summarised in table 3 and the self-report instruments used are described below.

### Fatigue severity

*Chalder Fatigue Questionnaire (CFQ)*[35] measures fatigue severity via 11 items scored against a 4-point Likert-type response scale. Scores are assigned for each response, using continuous scoring from 0 to 3. A cut-off of >18 defines a fatigue case.[35 36] Higher scores represent greater fatigue severity. The total score will be used here following recent psychometric evidence.[55 56] This scale displays

excellent psychometric properties[35 57] and has been validated among patients on HD.[56]

### Fatigue-related functional impairment

*Work and Social Adjustment Scale (WSAS)*[58] consists of five items that correspond to impairment in work, home management, social activities, private leisure activities and relationships as consequence of an illness or symptom, in this case fatigue. Higher scores indicate greater impairment. It has good psychometric properties[58] and has been previously used with patients on HD.[59]

### Depression

*Patient Health Questionnaire-9 (PHQ-9)*[60] measures depression over the last 2 weeks via a nine-item scale and an additional item to assess the impact of depression on functioning. Each item is scored from 0 (not at all) to 3 (nearly every day), with greater scores representing greater severity of depression. Depression severity cut-offs are available. The functional item is rated from 'Not at all difficult' to 'Extremely difficult'. The PHQ-9 displays excellent psychometric properties and is responsive to change.[61 62] The PHQ-9 has been validated in HD.[63]

### Anxiety

*Generalised Anxiety Disorder-7 (GAD-7)*[64] measures anxiety over 2 weeks, via a seven-item scale, and an additional item to assess the impact of anxiety on functioning. Each item is scored from 0 (not at all) to 3 (nearly every day), with greater scores representing greater severity of anxiety. Anxiety severity cut-offs are available. The functional item is rated from 'Not at all difficult' to 'Extremely difficult'. The PHQ-9 displays excellent psychometric properties.[64 65] This instrument has been used across chronic conditions, including HD patients.[42]

### Sleep quality

*Pittsburgh Sleep Quality Index (PSQI)*[66] measures seven components of sleep quality (subjective sleep quality, sleep latency, sleep duration, sleep efficiency, sleep disturbance, use of sleep medication and daytime dysfunction) over a 1 month time interval, via 19 items. Items are scored on an interval scale from 0 to 3. The scores of the components are then summed to obtain a global sleep quality score, ranging from 0 to 21. Higher scores indicate worse sleep quality. This scale displays satisfactory psychometric quality across patient populations[66 67] and is widely used and has been validated.[68]

### Process variables

#### Fatigue perceptions

*Brief Illness Perceptions Questionnaire (BIPQ)*[69] relies on a single-item approach to measure fatigue perceptions. It is a shorter version of the original Illness Perception Questionnaire (IPQ),[70] with moderate to good associations between the two.[69] Five of the items assess cognitive illness/symptom representations (consequences, timeline, personal control, treatment control and identity), two of them assess emotional representation (concern and emotions) and one item assesses illness/symptom comprehension. The items are rated using a response scale of 0–10. The psychometric properties of this measure have been assessed using samples from several illness groups, including renal disease,[69] displaying satisfactory quality.

### Cognitive and behavioural responses to fatigue

*Cognitive and Behavioural Responses to Symptoms Questionnaire (CBSQ)*[71] includes five cognitive subscales; fear avoidance, embarrassment avoidance, catastrophising about symptoms, beliefs that symptoms signal damage to the body (damage beliefs) and symptom focus, scored from strongly disagree (0) to strongly agree (4). There are also two behavioural subscales; resting and avoidance of activity and all-or-nothing behaviour. All items are scored on a five-point frequency scale ranging from never (0) to all the time (4). Item scores are added from each subscale to obtain a total score. Across studies, this instrument displays acceptable psychometric quality[72] and it has been used with different patient populations, including patients on HD.[59]

### Sleep hygiene behaviours

*Sleep Hygiene Index (SHI).*[73] measures sleep-related behaviours via 13 items. Each item is rated on a five-point scale, ranging from 0 (never) to 4 (always). Total scores range from 0 to 52, with higher scores representing poorer sleep hygiene. This scale displays adequate reliability and validity.[73 74] However, it has not yet been validated in kidney failure.

### Physical activity

*International Physical Activity Questionnaire–short form (IPAQ-SF).*[75] measures self-reported weekly time spent on physical activities (walks, physical exertion of moderate and vigorous intensities) and inactivity (sitting) via seven items. The questionnaire can be scored categorically according to developed cut-offs to classify individuals into low, moderate or high physical activity groups; or it can be scored continuously. Responses can be converted to metabolic equivalent task minutes per week (METmin/week), according to the IPAQ scoring protocol. MET scores across the three subcomponents can be summed to indicate overall physical activity.[75] The IPAQ is the most widely validated questionnaire, however, with some inconsistent evidence on its reliability and validity.[76] Given its brevity, simplicity and extensive use across research, it was selected here to measure physical activity.

### Adverse events

Information about occurrence of serious adverse events since the start of the study will be collected by self-report postintervention, according to good clinical practice guidelines. Adverse events will be flagged up to the trial management team and participants will be contacted to further assess the adverse event and its relationship to the study.

## Other treatments

Participants will be asked whether they have received any pharmacological, psychological or exercise-based treatment for depression and/or anxiety and/or fatigue in addition to BReF since starting the study.

## Demographic, social situational and clinical characteristics

At baseline, sociodemographic characteristics, including: gender, age, ethnicity, marital status, employment status, education, living arrangements, exercise, smoking status and alcohol consumption; and clinical characteristics, including: dialysis vintage and receipt of anaemia treatments will be collected via self-report.

Extra renal comorbidity will be assessed at baseline by consultant nephrologists using the Charlson Comorbidity Index (CCI).[77] This instrument is a weighted index that takes into account the number and the seriousness of comorbid diseases and can be adjusted for age. The method of classifying comorbidity provides a simple, readily applicable and valid method of estimating risk of death from comorbid disease for use in longitudinal studies.[77] The CCI has been previously used with and determined suitable for patients on dialysis.[78 79]

Clinical information, including: dialysis adequacy (urea reduction ratio), interdialytic weight gain, haemoglobin, ferritin, serum albumin, creatinine, urea, phosphate, potassium, calcium, C reactive protein and primary renal diagnosis will be extracted from patients' medical notes at baseline (T0) and postintervention (T1).

## Qualitative interviews

To complement the quantitative process evaluation and further explore the acceptability of the intervention, in line with current MRC process evaluation guidelines,[80] qualitative interviews will be conducted with a subgroup of participants from the intervention group at 3 months postrandomisation (T1). The interviews will be semi-structured and will be conducted over the phone or face-to-face, in a private environment. The interviews will be conducted by an independent researcher, who has not been involved in the intervention delivery. The main aim of the interviews will be to gather participants' experiences of the intervention, to identify areas of improvement. Purposive maximum variation sampling will be employed to ensure variability of the sample across a range of sociodemographic and clinical characteristics,[81] in particular: age, gender, ethnicity, dialysis vintage, degree of adherence to the intervention, degree of improvements in outcomes following the intervention. A minimum of 10 interviews will be conducted, until data saturation is reached, meaning the point where no new data is obtained with every new interview.[82]

## Sample size

The renal service of King's College Hospital has approximately 550 patients on HD and Lister Hospital has approximately 510 patients on HD, in which we expect to be able to approach 636 (60%) during the recruitment period. Past psychological research in patients on dialysis, conducted by the team, suggest consent rates between 50% and 70%, assuming a more conservative uptake of 40%, 254 patients are expected to be screened, with approximately 30% (n=76) expected to meet the inclusion criteria, including around half reporting clinical levels of fatigue.[59] Assuming 50% of those eligible will consent to be randomised (n=38), a sample size of 40 participants would allow us to estimate the true population consent rate with a 11% margin of error (95% binomial exact confidence level) for those meeting eligibility criteria. In line with recommended sample sizes of pilot feasibility trials,[83–85] 40 patients is deemed sufficient to explore feasibility, acceptability and potentially efficacy of the intervention, assuming retention rates of 80%, the true population consent rate will be with a margin of error of 13% (95% binomial exact CI), an acceptable level of error, based on the time, budget and workforce constraints, as FP will act as both the trial coordinator and therapist.

## Analysis plan

Descriptive statistics of patients approached, screened, eligible, consented and randomised will be computed to address objectives 1–2. Reasons for non-consent, exclusion and drop-out, at each stage of the study, will be recorded and reported. Adherence to the intervention will be reported using descriptive statistics to address objective 3. The following values will be computed: mean number of homework tasks completed, mean number of sessions completed, a breakdown on the number of participants completing each session, number of participants completing the independent chapter, mean duration of the telephone and face-to-face sessions. SD by trial arm will be computed for the fatigue outcomes in order to estimate a more robust sample size for a future efficacy trial, thereby, addressing objective 4.

The psychometric quality of the self-report instruments used will be assessed to address objective 5. Reliability will be assessed using Cronbach's alpha, with a minimum acceptable cut-off at $\alpha=0.70$, but preferably at $\alpha=0.80$ or higher, particularly for the key variables[86] and individual items will be checked to ensure that there are no problematic items for this patient population. Convergent validity will be assessed via Pearson's correlations between psychological constructs (eg, depression and fatigue severity) and clinical markers (eg, comorbidities and fatigue severity). Additionally, content validity of the fatigue measures will be considered based on the qualitative data to ensure that the selected measures capture changes described by participants. Responsiveness/sensitivity to change will be assessed by checking correlations between change scores on key variables of interest and by triangulation with participants' narratives.

Given the feasibility nature of the trial, statistical significance will not be assessed; instead effect sizes and CI will be estimated. An analysis of covariance (ANCOVA) will be performed to estimate the postintervention

mean difference in outcomes, controlling for the baseline levels of each outcome, for the following variables: fatigue severity, fatigue-related functional impairment, depression, anxiety and subjective sleep quality, thereby addressing objectives 6 and 7. Group allocation will be included as an indicator variable following the intention-to-treat principle. Recruitment centre will also be controlled for in the analysis as it is a stratification factor. Differences in intervention effects by sociodemographic and clinical factors on fatigue outcomes will be explored.

Changes in fatigue perceptions, and cognitions and behaviours in response to fatigue will be evaluated via ANCOVAs to address objective 8. The proportion of the treatment effect that may be accounted for by these process variables with CI will also be calculated, as there will be insufficient power for mediation analyses.

To meet objective 9 and qualitatively explore the acceptability and usefulness of the intervention from the perspective of the participants, the semistructured qualitative interviews will be transcribed verbatim and analysed using inductive thematic analysis with the use of NVivo software. Thematic analysis revolves around identifying recurrent themes and patterns from the interviews and developing a coding manual.[87]

To address objective 10, a mixed methods approach will be used, drawing on both the quantitative and qualitative findings to determine any intervention-specific issues, such as the optimal number of sessions.

Considering issues relating to recruitment and retention rates, suitability of the selected measures, as well as any intervention-specific issues, will help us to determine whether to proceed to a full-scale efficacy trial, if so, these findings will also inform aspects of the design of the efficacy trial, such as the required sample size and appropriate self-report measures to ensure sufficient power and sensitivity to detect any intervention effects.

## DISCUSSION

Fatigue is common in chronic HD patients with consequences on patients' functioning and daily living, as well as implications on clinical outcomes. BReF is a theory-driven and evidence-based CBT intervention with therapist support aimed at improving renal fatigue, that has been designed following the Medical Research Council guidance for developing and evaluating complex interventions.[88]

This is the first feasibility RCT to examine whether a fatigue-specific CBT-based programme with therapist support is feasible, acceptable and possibly beneficial at reducing fatigue severity and fatigue-related functional impairment in patients undergoing in-centre HD who are fatigued. The brief and stepped structure of the intervention may be more appropriate for patients receiving in-centre HD. Prior to proceeding to a full-scale trial, it is important to identify unique challenges with recruitment and retention in this particular setting and to explore whether the content and structure of the manual

are deemed useful and relevant by patients. The results of the BReF trial will inform the design of a future full-scale trial powered to detect the efficacy of CBT for the management of fatigue in HD, accompanied by a longer follow-up to assess any sustained effects of the intervention on outcomes.

### Ethics

The trial is co-sponsored by King's College London and King's College Hospital NHS Foundation Trust. Patient consent will be obtained. The Chief Investigator (CI) and all members of the research team will preserve the confidentiality of participants taking part in the study and will work in accordance with the Caldicott Principles, Data Protection Act 1998, NHS Code of Confidentiality and any relevant NHS Trust organisational policies. All serious adverse events related to the study will be reported to the study sponsor, ethics committee and relevant NHS R&D departments. Authorisation will be sought from the study sponsor for any future substantial and non-substantial amendments arising during the course of the study, prior to submission to the HRA. The study may be subject to inspection and audit by King's College London under their remit as sponsor and other regulatory bodies to ensure adherence to GCP and the NHS Research Governance Framework for Health and Social Care (second edition).

### Indemnity

The study is cosponsored by King's College London and King's College Hospital, providing insurance for the study, through its own professional indemnity for research involving human participants and no fault compensation and the Trust having a duty of care to patients via NHS indemnity cover, in respect of any claims arising as a result of clinical negligence by its employees, brought by or on behalf of a study patient.

### Dissemination

We will endeavour to publish the findings of this trial in a peer-reviewed journal and present the findings at relevant national and international conferences.

### Trial status

The study will start recruitment at the end of October 2017. Recruitment will continue until July 2018. Patient involvement in the study will conclude in November 2018.

**Contributors** All authors contributed to the design of the study and writing of the protocol. FP, RM-M, JC and ICM were involved in the development of the BReF intervention. FP, JC and SN were involved in the statistical analysis plan. The day-to-day management of the study will be co-ordinated by the CI and the trial management team. The team will also meet regularly, once a month, to discuss the overall running of the study, including rates of recruitment, adherence to the protocol and safety of patients.

**Funding** This work is embedded within a larger PhD project funded by a Biomedical Research Studentship to Miss Federica Picariello from the National Institute for Health Research (NIHR) Biomedical Research Centre at South London and Maudsley NHS Foundation Trust and King's College London. The views expressed are those of the authors and not necessarily those of the NHS, the NIHR or the Department of Health.

**Competing interests** None declared.

**Ethics approval** London Bridge NHS Research Ethics Committee (17/L0/1406).

**Provenance and peer review** Not commissioned; externally peer reviewed.

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
