## [Reviewer comments · BMJ Open]

ARTICLE DETAILS

TITLE (PROVISIONAL)	Cognitive Behavioural Therapy (CBT) for Renal Fatigue (BReF): A feasibility randomised-controlled trial of CBT for the management of fatigue in haemodialysis (HD) patients.
AUTHORS	Picariello, Federica; Moss-Morris, Rona; Macdougall, Iain; Norton, Sam; Da Silva-Gane, Maria; Farrington, Ken; Clayton, Hope; Chilcot, Joseph

VERSION 1 – REVIEW

REVIEWER	Ikechi Okpechi University of Cape Town South Africa
REVIEW RETURNED	17-Dec-2017

GENERAL COMMENTS	A detailed and well designed study by Picariello et al. One point I would like them to address regarding inclusion / exclusion of patients to this study is that of anaemia causing fatigue in HD patients. As anaemia is one of the major (known) causes of fatigue in this population, will patients with anaemia be excluded? If so, at what level of haemoglobin will patients be allowed to take part in the study? Otherwise, the authors should clarify why anaemia will not be a consideration for inclusion / exclusion.
---

REVIEWER	Zhiguo Mao Division of Nephrology Kidney Institute of CPLA Changzheng Hospital Second Military Medical University Shanghai, China
REVIEW RETURNED	19-Dec-2017

GENERAL COMMENTS	This is a well-designed and clearly-described protocol. The methodology is appropriate and there are no ethics concerns in this study.
--

VERSION 1 – AUTHOR RESPONSE

First of all, we would like to thank both reviewers for their time and kind feedback.

We have addressed the comment raised by Professor Okpechi. The changes are marked in the manuscript in red and detailed below.

Professor Okpechi, thank you for raising this to our attention. Patients will not be screened for anaemia as part of this feasibility trial.

As you rightly pointed out, there is little question that severe anaemia can cause fatigue; however, when levels of haemoglobin and haematocrit are maintained within recommended ranges, the link with fatigue is no longer present. With improvements in clinical care, most patients are very well controlled biochemically. Additionally, there is evidence that the greatest improvements in fatigue are evident among patients whose baseline haemoglobin is partially corrected to a minimum of ≥ 10 g/dL, but not among those with full correction to ≥ 12 g/dL, suggesting some form of ceiling effect and improvements in fatigue with management of anaemia are often below a clinically meaningful threshold, particularly in dialysis patients. These are the reasons why in this feasibility trial, we decided not to screen patients for anaemia. However, we will evaluate this in the exploratory analysis and we will revise the inclusion/exclusion criteria accordingly for a future efficacy trial. This has been added in red to the manuscript (please see page 7).